# Évaluer les tableaux de revue de littérature générés par les LLM : état de l'art et étude préliminaire

Marah Baccari[1]     Florian Boudin[2]     Richard Dufour[1]
[1]Nantes Université, École Centrale Nantes, CNRS, LS2N, UMR 6004, F-44000 Nantes, France
[2]Inria, LS2N, Nantes Université, France
`prénom.nom@univ-nantes.fr`

## RÉSUMÉ

Cet article dresse un état de l'art des méthodes de génération automatique de tableaux de revue de littérature à l'aide de grands modèles de langue (LLM), ainsi que des approches proposées pour évaluer leur qualité. Les tableaux de synthèse constituent un format privilégié pour structurer et comparer des travaux scientifiques, mais leur génération automatique soulève des défis importants en termes d'exactitude, de cohérence et de complétude. Nous passons en revue les principales approches existantes et les stratégies d'évaluation associées, en mettant en évidence leurs limites, notamment la dépendance à des références de comparaison et l'insuffisance des métriques actuelles pour capturer la qualité globale des tableaux. Enfin, nous proposons des pistes pour une évaluation plus robuste, en introduisant différentes dimensions d'analyse et en explorant des stratégies alternatives. Cet article constitue ainsi une première étape vers l'élaboration d'un cadre d'évaluation plus adapté aux tableaux de revue de littérature générés automatiquement.

## ABSTRACT

### Evaluating LLM-Generated Literature Review Tables : A Survey and Preliminary Study

This article presents a state-of-the-art review of methods for automatically generating literature-review tables using large language models (LLMs), as well as the approaches proposed to evaluate their quality. Summary tables are a key format for organizing and comparing scientific work, but their automatic generation raises important challenges related to accuracy, coherence, and completeness. We examine the main existing approaches and their associated evaluation strategies, highlighting their limitations, including their dependence on reference tables and the inability of current metrics to capture the overall quality of generated tables. Finally, we outline directions for more robust evaluation, introducing multiple analytical dimensions and exploring alternative strategies. Thus, this article represents a first step toward developing a more suitable evaluation framework for automatically generated literature-review tables.

MOTS-CLÉS : Tableaux de revue de littérature, LLM, Evaluation.

KEYWORDS: Literature-review tables, LLMs, Evaluation.

# 1   Introduction

La croissance rapide du nombre de publications scientifiques rend l'accès à la littérature, sa compréhension et sa synthèse de plus en plus difficiles. Les chercheurs font face à une surcharge informationnelle qui complique la réalisation de revues de littérature, pourtant essentielles pour situer un travail dans

son domaine et identifier les avancées existantes.

Dans ce contexte, les tableaux de synthèse jouent un rôle important pour organiser et comparer les travaux antérieurs (Webster & Watson, 2002; Schünemann *et al.*, 2008; Younas & Ali, 2021). En représentant les articles en lignes et les critères de comparaison en colonnes, ils offrent une vue d'ensemble structurée qui facilite l'analyse comparative, la mise en évidence de tendances et l'identification de lacunes dans la littérature. Cependant, leur élaboration demeure une tâche coûteuse, nécessitant la lecture de nombreux articles, l'identification de dimensions de comparaison pertinentes, ainsi que l'extraction et l'organisation cohérentes des informations.

Cette difficulté est particulièrement marquée lors de la définition du schéma du tableau, c'est-à-dire le choix des colonnes. Contrairement au remplissage des cellules, cette étape exige une compréhension globale du domaine ainsi que la capacité à faire émerger des critères de comparaison implicites, souvent absents ou dispersés dans les articles (Padmakumar *et al.*, 2025). Cette tâche reste encore peu étudiée, alors même qu'elle constitue un obstacle majeur à l'automatisation du processus.

Les grands modèles de langue (LLM) apparaissent dès lors comme des outils prometteurs pour automatiser ou assister la génération de tableaux de revue de littérature. Grâce à leurs capacités d'extraction, de structuration et de synthèse, ils ouvrent de nouvelles perspectives pour accélérer la production de ces tableaux et en faciliter l'usage. Plusieurs travaux récents se sont ainsi intéressés à l'utilisation des LLM pour générer des synthèses structurées à partir d'articles scientifiques (Newman *et al.*, 2024; Padmakumar *et al.*, 2025; Wang *et al.*, 2025).

Cette automatisation soulève toutefois plusieurs questions à différents niveaux. En amont, la définition même de la tâche reste ouverte : comment formuler le besoin d'information et l'instruction adressée au modèle, et quels types d'entrées considérer ? Au niveau de la génération, se posent notamment les questions de la construction du schéma du tableau et du remplissage des cellules. En aval, l'évaluation des tableaux générés constitue un défi majeur : comment en mesurer la qualité de manière fiable, au-delà de simples correspondances de surface ?

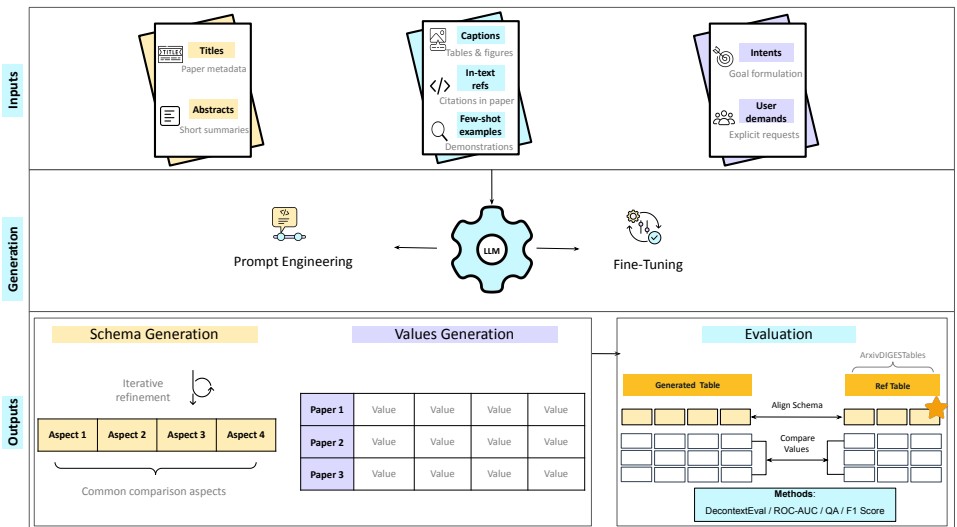

FIGURE 1 – Vue d'ensemble du pipeline de génération et d'évaluation de tables

Plusieurs travaux récents ont commencé à structurer ce problème (Figure 1). En particulier, Newman *et al.* (2024) proposent ArxivDIGESTables, un premier jeu de données composé de tableaux issus de revues de littérature disponibles sur arXiv, ainsi qu'une première formalisation de la tâche. La Figure 1 illustre ce cadre à travers un pipeline typique : à partir d'un ensemble d'articles scientifiques, un LLM génère un tableau de synthèse structurant les informations extraites.

Dans la continuité de ce travail, plusieurs études se sont intéressées à des aspects spécifiques. Certaines visent à enrichir les jeux de données, notamment en introduisant des intentions ou des formulations explicites du besoin d'information, sous forme de questions ouvertes ou de descriptions abstraites, afin de guider la génération (Padmakumar *et al.*, 2025; Wang *et al.*, 2025). D'autres explorent des méthodes pour améliorer la qualité des schémas produits, notamment via des approches de raffinement post-génération (Padmakumar *et al.*, 2025), ou des stratégies itératives de raffinement permettant d'ajuster progressivement le schéma au fil de plusieurs passes sur les documents (Wang *et al.*, 2025).

Ces travaux proposent également des approches d'évaluation plus fines, motivées par les limites des métriques actuelles, souvent fondées sur des représentations statiques incapables de saisir des variations sémantiques contextuelles (Wang *et al.*, 2025). Des alternatives incluent la décontextualisation des colonnes et l'évaluation par ROC-AUC pour mesurer l'alignement des schémas (Padmakumar *et al.*, 2025), ou encore des protocoles fondés sur des questions-réponses synthétiques pour évaluer la préservation du schéma, des valeurs et des relations entre cellules (Wang *et al.*, 2025).

L'évaluation du schéma apparaît ainsi comme un enjeu particulièrement déterminant. Alors que le remplissage des cellules peut être rapproché de tâches d'extraction (Wu *et al.*, 2022) ou de question-réponse (Du *et al.*, 2021) relativement bien étudiées, la construction du schéma implique des choix de structuration plus abstraits, difficiles à formaliser et à évaluer automatiquement (Padmakumar *et al.*, 2025). L'absence de cadre d'évaluation adapté, prenant en compte à la fois la structure et le contenu des tableaux, constitue un frein important au développement et à l'adoption de ces approches.

Dans cet article, nous proposons un état de l'art des travaux existants sur la génération de tableaux de revue de littérature à l'aide des LLMs, avec une attention particulière sur l'évaluation. Nous analysons les approches proposées, mettons en évidence leurs limites et discutons des défis ouverts. Enfin, nous esquissons des pistes pour une évaluation plus robuste, constituant une première étape vers un cadre méthodologique adapté.

# 2 Automatisation des tableaux de revue de littérature

## 2.1 Formulation de la tâche

La génération d'un tableau de revue de littérature peut être vue comme une tâche de structuration d'information à partir de documents scientifiques longs, hétérogènes et riches en contenu. Étant donné un ensemble de $M$ articles, l'objectif est de produire un tableau composé de $M$ lignes, chacune correspondant à un article, ainsi que d'un ensemble de colonnes définissant un schéma de comparaison. Chaque cellule du tableau contient une valeur extraite des documents sources.

Dans ce cadre, les travaux existants proposent différentes formulations de la tâche, qui se distinguent par le degré de structuration du processus de génération ainsi que par les types d'entrées utilisés pour guider les modèles.

Dans le travail de Newman *et al.* (2024), la tâche est formulée comme une tâche de reconstruction de tableaux de revue de littérature à partir d'un ensemble d'articles scientifiques. Les auteurs adoptent une approche en deux étapes, en distinguant la génération du schéma de celle des valeurs. Pour la construction du schéma, ils explorent différentes configurations d'entrées, allant des métadonnées minimales (titres et résumés) à des contextes plus riches incluant des légendes générées automatiquement, des légendes issues des articles, ainsi que des références dans le texte mentionnant explicitement les tableaux. Ils évaluent également l'apport d'exemples en contexte (*few-shot*), récupérés par similarité sémantique, afin de guider la génération du schéma. La génération des valeurs repose quant à elle sur le texte intégral des articles, qui fournit les informations nécessaires pour remplir les cellules du tableau. Les expériences menées montrent que les entrées enrichies, notamment les références dans le texte et les exemples en contexte, permettent d'améliorer la couverture des aspects du schéma, tandis que le texte intégral est indispensable pour produire des valeurs précises.

Padmakumar *et al.* (2025) se concentrent davantage sur la génération du schéma. Partant de l'idée que plusieurs schémas peuvent être valides, ils proposent une reformulation de la tâche en introduisant explicitement l'objectif du tableau à travers la notion de *table intent*, générée à l'aide d'un LLM à partir des titres et résumés des documents. Les auteurs considèrent que la génération d'un schéma est intrinsèquement ambiguë en l'absence d'une description claire du besoin informationnel. La tâche consiste alors, à partir d'un ensemble de documents et d'une intention, à générer un schéma structuré composé d'aspects enrichis, incluant non seulement des noms de colonne mais également des définitions et des formats attendus. Les auteurs évaluent différentes configurations d'entrée, combinant les titres et résumés avec des légendes, des références textuelles, ou encore le texte intégral, ainsi que des exemples en contexte. Ils étudient également des stratégies séquentielles de génération et de révision du schéma, dans lesquelles des schémas candidats sont progressivement améliorés à l'aide de mécanismes de guidage ou de critique. Les résultats montrent que l'intégration d'une intention améliore significativement la qualité des schémas générés, tandis que l'utilisation du texte intégral à cette étape apparaît moins efficace.

Enfin, Wang *et al.* (2025) étendent la formulation de la tâche en proposant un pipeline complet couvrant à la fois la sélection des documents et la génération du tableau. Leur approche se décompose en trois étapes : 1) la récupération de documents candidats, 2) leur sélection en présence d'articles distracteurs, puis 3) la génération du tableau. Dans ce cadre, de la même façon que (Padmakumar *et al.*, 2025), la génération du schéma et des valeurs est guidée par une description explicite du besoin, appelée *user demand*, qui remplace les légendes jugées trop vagues. Les auteurs utilisent les titres et résumés pour représenter les articles lors des étapes de récupération et de sélection, tandis que le texte intégral est exploité sous une forme condensée, par extraction des informations clés, afin de limiter la taille du contexte. La génération du tableau est réalisée de manière itérative, en traitant les articles par groupes successifs, ce qui permet d'affiner progressivement le schéma et les valeurs produites. Les expériences montrent que l'utilisation de *user demands* améliore l'adéquation du tableau aux besoins exprimés, et que la génération itérative contribue à améliorer la qualité globale du résultat.

Ces travaux mettent ainsi en évidence l'importance conjointe de la formulation de la tâche et du choix des entrées pour la génération de tableaux de revue de littérature. Ils montrent également que différentes stratégies peuvent être adoptées selon les objectifs visés, qu'il s'agisse de reproduire fidèlement un tableau existant ou de générer un tableau adapté à un besoin informationnel spécifique.

## 2.2 Jeux de données

Les travaux récents sur la génération de tableaux de revue de littérature reposent en grande majorité sur un même jeu de données de référence, introduit par Newman *et al.* (2024), puis repris dans les travaux ultérieurs sous différentes formes d'enrichissement. Plutôt que de proposer de nouveaux corpus entièrement distincts, les contributions successives consistent principalement à adapter cette base initiale afin de mieux refléter certains aspects de la tâche, tels que l'expression de l'intention ou le réalisme des scénarios d'utilisation. Le tableau 1 synthétise les principales caractéristiques de ces différentes versions.

| Jeu de données | #Tables | #Articles | Description | Intention |
|---|---|---|---|---|
| ArxivDIGESTables (Newman *et al.*, 2024) | 2 228 | 7 542 | Tables réelles extraites automatiquement à partir d'articles arXiv | Non |
| ArxivDIGESTables-with-intent (Padmakumar *et al.*, 2025) | 2 228 | 7 542 | Ajout d'intentions générées automatiquement | Oui (table intent) |
| ArxivDIGESTables-Clean (Padmakumar *et al.*, 2025) | 100 | – | Sélection d'un sous-ensemble nettoyé et annoté manuellement | Oui (table intent) |
| ARXIV2TABLE (Wang *et al.*, 2025) | 1 957 | 7 158 | Filtrage du corpus initial, ajout de distracteurs et formulation des besoins via user demands | Oui (user demand) |

TABLE 1 – Comparaison des principaux jeux de données de tableaux de revue de littérature

Comme mentionné précédemment, le jeu de données ArxivDIGESTables est construit à partir de tableaux extraits automatiquement de sources LaTeX issues de publications arXiv [1]. Le processus de construction repose sur plusieurs étapes de filtrage visant à garantir la qualité des données : suppression des tables mal parsées, exclusion des tableaux trop petits ou trop volumineux, élimination des tables sans citations explicites, ainsi que des colonnes non textuelles. Une vérification manuelle finale est également effectuée. Le corpus obtenu contient 2 228 tableaux, correspondant à 11 016 lignes (issues de 7 542 articles distincts) et 7 634 colonnes, pour un total de 43 905 valeurs. L'analyse des dimensions des tableaux montre une forte variabilité : le nombre de lignes varie entre 1 et 35, tandis que le nombre de colonnes varie entre 2 et 13, ce qui reflète la diversité des structures observées dans les revues de littérature.

Une analyse plus fine du contenu met en évidence la nature des informations représentées. Environ 40 % des colonnes sont de type catégoriel ou booléen, ce qui les rend particulièrement adaptées à la comparaison entre articles, tandis que les 60 % restants correspondent à des contenus descriptifs. Les thématiques des colonnes sont également variées : une annotation manuelle réalisée par Newman *et al.* (2024) sur un sous-ensemble de tableaux montre que 38 % des colonnes concernent les jeux de données, 20 % les méthodes, et le reste couvre des aspects tels que les tâches ou les applications. Le corpus est toutefois fortement déséquilibré du point de vue disciplinaire : plus de 90 % des tableaux (1 985 articles) proviennent du domaine de l'informatique, tandis que les autres domaines, comme la physique, les mathématiques ou la biologie, sont faiblement représentés. Chaque instance est structurée sous format JSON, incluant non seulement le tableau lui-même (colonnes et valeurs), mais aussi des métadonnées telles que l'identifiant du tableau, la légende, les références dans le texte (*in-text references*) et les informations associées aux articles cités (titres, résumés et texte intégral). Par exemple, les références textuelles permettent de localiser précisément les passages décrivant le tableau dans l'article source, ce qui constitue un signal important pour la génération.

---

1. https://arxiv.org/

Notre analyse de la distribution des colonnes dans *ArxivDIGESTables* met en évidence une forte hétérogénéité des schémas, malgré la présence de certaines dimensions récurrentes. La colonne la plus fréquente, *year*, n'apparaît que dans 11,5 % des tableaux (256 occurrences), suivie de *dataset* (6,6 %), *task* (5,9 %) et *category* (5,4 %). Les autres colonnes courantes, telles que *method*, *model* ou *description*, restent également peu dominantes (moins de 5 %). Ces résultats montrent qu'il n'existe pas de schéma standard partagé à grande échelle. Cette dispersion confirme la variabilité structurelle des tableaux et souligne la difficulté de la tâche de génération, qui nécessite d'inférer dynamiquement des schémas adaptés à chaque contexte plutôt que de s'appuyer sur un ensemble fixe de colonnes.

Sur la base de ce corpus, Padmakumar *et al.* (2025) proposent une première extension visant à expliciter le besoin informationnel associé à chaque tableau à travers la notion d'*intent*. Pour chaque instance, une intention est générée automatiquement à l'aide d'un LLM (GPT-4o), à partir de la légende, des références dans le texte et des métadonnées des articles. Ce processus repose sur la génération de 5 intentions candidates, puis sur la sélection de la meilleure à l'aide d'un juge LLM. Notre analyse quantitative montre que ces intentions sont relativement courtes (en moyenne 25 tokens), mais significativement plus informatives que les légendes seules (en moyenne 16 tokens), tout en restant très concises par rapport aux résumés des articles (plus de 800 tokens en moyenne). En parallèle, les auteurs construisent un sous-ensemble annoté manuellement, ArxivDIGESTables-Clean, en sélectionnant initialement des tableaux respectant certains critères de taille (au moins 5 lignes et 4 colonnes), puis en supprimant ceux dont plus de 50 % des colonnes sont jugées trop génériques ou trop spécifiques et qui traitent finalement les cas de fuite d'informations du tableau vers la légende. Après filtrage, 100 tableaux sont retenus et entièrement vérifiés par des annotateurs humains. Une particularité de cette version enrichie est l'ajout, pour chaque colonne, d'une définition textuelle et d'un format attendu, ce qui permet de mieux caractériser les aspects du schéma et de faciliter leur génération.

Enfin, Wang *et al.* (2025) proposent une adaptation du corpus visant à se rapprocher de scénarios d'utilisation réels, à travers le jeu de données ARXIV2TABLE. Dans un premier temps, les auteurs construisent ce corpus à partir de ArxivDIGESTables en appliquant un filtrage des tableaux jugés structurellement incomplets. Ce processus vise à garantir la cohérence des données et la disponibilité des informations nécessaires aux différentes étapes du pipeline. Après ce filtrage, le nombre de tableaux est réduit de 2 228 à 1 957, couvrant 7 158 articles.

Contrairement aux travaux précédents, le jeu de données reconstruit par Wang *et al.* (2025) est présenté par les auteurs comme plus réaliste, dans la mesure où il simule un véritable processus de revue de littérature : formulation d'un besoin *user demand*, recherche de papiers, filtrage des résultats, gestion des distracteurs et enfin la génération progressive du tableau. En effet, chaque instance inclut un ensemble de papiers candidats contenant des *distracteurs*, c'est-à-dire des articles proches sémantiquement mais non pertinents. Ces documents sont d'abord récupérés automatiquement à l'aide de représentations vectorielles basées sur les titres et résumés, puis filtrés par un modèle de langage et validés par annotation humaine. Cette construction permet d'évaluer non seulement la génération du tableau, mais également la capacité des modèles à sélectionner les documents pertinents dans un contexte bruité.

La génération du tableau est ensuite réalisée de manière itérative, en traitant les articles par groupes successifs, ce qui permet d'ajuster progressivement le schéma. En outre, de la même manière que Padmakumar *et al.* (2025), les légendes des tableaux sont remplacées par des *user demands*, jouant un rôle similaire à celui des *intents*. Ces descriptions sont significativement plus longues que les intentions (environ 49 tokens en moyenne) et plus détaillées que les légendes initiales (environ 16 tokens).

Cette différence reflète une volonté de formuler des requêtes plus proches de celles exprimées par un utilisateur réel. Par exemple, une légende telle que « *Summary of SWIPT Beamforming Designs* » peut être reformulée en une demande utilisateur décrivant explicitement le besoin de l'utilisateur, dans un esprit proche des *intents*. La Figure 2 illustre la différence entre légende, intent et user demand. Ces formulations visent ainsi à mieux guider la génération du schéma.

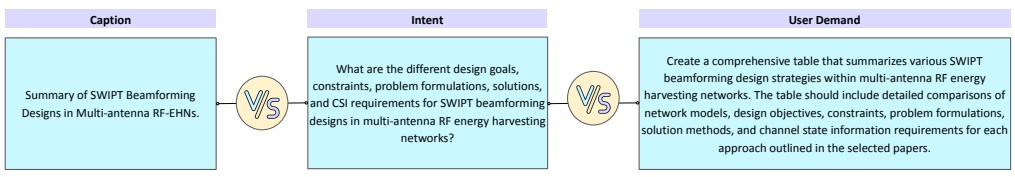

FIGURE 2 – Exemple comparatif entre légende, intention et demande utilisateur.

## 2.3 Evaluation

Dans la continuité des différentes formulations de la tâche et des jeux de données présentés précédemment, la question de l'évaluation constitue un enjeu central pour les travaux sur la génération de tableaux de revue de littérature. En effet, la diversité des entrées, des objectifs et des scénarios d'utilisation rend difficile la définition de protocoles d'évaluation à la fois fiables, comparables et représentatifs des usages réels. De manière générale, l'évaluation consiste à comparer un tableau généré à un tableau de référence, comme illustré dans la Figure 3. Toutefois, cette comparaison est rendue complexe par des différences fréquentes de structure, de granularité et de formulation. Des colonnes sémantiquement équivalentes peuvent être exprimées différemment, certaines informations peuvent être absentes ou, au contraire, plus détaillées, et plusieurs schémas valides peuvent exister pour un même ensemble de documents (Padmakumar *et al.*, 2025). Ces éléments limitent fortement la portée des métriques classiques fondées sur des correspondances de surface, telles que l'égalité exacte ou les similarités lexicales simples, qui capturent mal les relations sémantiques entre colonnes et valeurs.

| | Dataset | Size | Task | Metric |
|---|---|---|---|---|
| **Paper 1** | ImageNet | 1500 | Classification | Precision |
| **Paper 2** | CIFAR | 682 | Classif. | - |
| **Paper 3** | ImageNet | 120k | Labeling | F1 |

Generated Table

- ☐ Columns only in generated table
- ☐ Same columns with different names
- ☐ Matching values
- ☐ Columns only in reference table
- ☐ Columns only in reference table

| | Application | Data | Method |
|---|---|---|---|
| **Paper 1** | Classif. | 1,500 images | CNN |
| **Paper 2** | Classif. | 672 mages | Transformers |
| **Paper 3** | Labeling | 120.000 | GNN |

Reference Table

FIGURE 3 – Alignement d'un tableau de revue de littérature généré avec un tableau de référence

Dans ce contexte, Newman *et al.* (2024) proposent un cadre d'évaluation qui dissocie le schéma et les valeurs, afin de distinguer les erreurs structurelles des erreurs de contenu. Leur contribution principale est *DecontextEval*, une méthode d'évaluation automatique conçue pour prendre en compte les variations sémantiques entre colonnes. Cette approche repose d'abord sur une étape de décontextualisation, dans laquelle les intitulés de colonnes sont reformulés en descriptions explicites et indépendantes du tableau, à l'aide d'un modèle de langage. Ces descriptions sont ensuite encodées sous forme de représentations vectorielles, et la similarité entre colonnes générées et colonnes de

référence est mesurée par similarité cosinus. Un appariement optimal entre colonnes est alors réalisé, permettant de calculer des métriques classiques de précision, rappel et F1 pour le schéma. Une approche similaire est utilisée pour l'évaluation des valeurs, en comparant les cellules alignées après l'alignement des colonnes. Les auteurs évaluent également l'impact de différentes entrées (titres, résumés, légendes, références textuelles, exemples en contexte) sur ces métriques. Les résultats montrent que des entrées riches en contexte améliorent significativement le rappel des colonnes, mais peuvent réduire la précision en raison de l'ajout de colonnes supplémentaires. Par ailleurs, les auteurs complètent cette évaluation automatique par une étude humaine, qui met en évidence que certaines colonnes générées, bien qu'absentes du tableau de référence, peuvent tout de même être jugées utiles dans le cadre d'une revue de littérature.

Dans la continuité de ces travaux, Padmakumar *et al.* (2025) se concentrent exclusivement sur la qualité du schéma généré. Les auteurs adoptent un cadre d'alignement similaire à celui de Newman *et al.* (2024) : chaque colonne générée est comparée à chaque colonne de référence à l'aide de BERTScore (Zhang *et al.*, 2020), calculé sur la concaténation du nom, de la définition et du format attendu de l'aspect. Une paire de colonnes est considérée comme appariée lorsque la similarité dépasse un seuil, traité comme un hyperparamètre de l'évaluation. Pour éviter de choisir arbitrairement un seuil unique, ils calculent la précision, le rappel et le F1 pour une série de seuils allant de 0,40 à 1,00, avec un incrément de 0,01, puis rapportent pour chaque métrique l'aire sous la courbe (AUC) obtenue via la règle du trapèze. Selon les auteurs, cette stratégie permet de capturer l'évolution complète des performances du modèle sur l'ensemble des seuils possibles, offrant une mesure plus robuste que l'utilisation d'un seuil fixe. Les résultats montrent que l'intégration explicite de l'*intent* améliore significativement la pertinence des schémas générés.

Enfin, Wang *et al.* (2025) proposent une méthode d'évaluation par questions-réponses (QA), où des paires de questions sont automatiquement synthétisées à partir du tableau de référence, puis posées au tableau généré, et inversement. Cette stratégie permet d'évaluer trois dimensions essentielles de la qualité d'un tableau : (1) le schéma, en vérifiant si une colonne donnée du tableau de référence est bien présente dans le tableau généré ; (2) les valeurs unitaires, en testant si une cellule spécifique du tableau d'origine apparaît dans le tableau produit ; (3) les relations binaires, en examinant si les relations entre deux cellules (par exemple, un ordre, une co-occurrence ou une association) sont préservées. Pour mesurer le rappel, GPT-4o (OpenAI *et al.*, 2024) génère des questions binaires ("oui/non") à partir du tableau de référence : toutes les colonnes et toutes les cellules donnent lieu à des questions, tandis que dix paires de cellules sont échantillonnées pour les relations binaires. Le modèle doit répondre à ces questions en s'appuyant uniquement sur le tableau généré ; lorsqu'une information est absente, il doit répondre "non". Le pourcentage de réponses "oui", mesurant le rappel, indique alors la proportion d'informations du tableau d'origine effectivement retrouvées dans la génération. Pour mesurer la précision, le processus est inversé : les questions sont synthétisées à partir du tableau généré, puis un autre LLM doit y répondre en consultant cette fois le tableau de référence.

Dans l'ensemble, ces travaux mettent en évidence une évolution progressive des stratégies d'évaluation. Les premières approches reposent sur des mesures de similarité entre schémas et valeurs, enrichies par des techniques d'alignement sémantique. Elles sont ensuite étendues pour intégrer explicitement le besoin informationnel à travers la notion d'*intent*, permettant de réduire l'ambiguïté de la tâche. Enfin, des approches plus récentes proposent des protocoles orientés *usage*, évaluant directement la capacité des tableaux à répondre à des besoins concrets. Malgré ces avancées, l'évaluation reste un problème ouvert, notamment en raison de la difficulté à concilier fidélité structurelle, pertinence sémantique et utilité réelle des tableaux générés.

# 3 Limites et perspectives

Malgré les avancées proposées dans les travaux récents, plusieurs limites importantes subsistent et soulignent la complexité intrinsèque de la génération et de l'évaluation des tableaux de revue de littérature. Un premier verrou majeur concerne les jeux de données. La plupart des travaux existants s'appuient sur une même source, *ArxivDigestTables* (Newman *et al.*, 2024), souvent réutilisée après nettoyage, filtrage ou transformation. Cette dépendance à une unique source limite fortement la diversité des données, d'autant plus que celles-ci proviennent majoritairement du domaine de l'informatique. Par conséquent, la généralisation des approches à d'autres disciplines reste incertaine.

Une autre limite concerne la présence d'indices explicites sur le schéma cible dans les entrées textuelles utilisées pour guider la génération, notamment les *intents* introduits par Padmakumar *et al.* (2025) et les *user demands* proposés par Wang *et al.* (2025). En effet, bien que ces formulations soient présentées comme des descriptions de besoins informationnels, elles contiennent fréquemment des mentions directes des colonnes attendues, ce qui réduit artificiellement la difficulté de la tâche en fournissant au modèle une partie du schéma cible.

Notre analyse quantitative met clairement en évidence ce phénomène. Dans le jeu de données *ArxivDIGESTables-with-intent* (Padmakumar *et al.*, 2025), 49,2 % des tableaux contiennent au moins une colonne explicitement mentionnée dans l'*intent* (1 096 tables), et 7,4 % des cas présentent une correspondance complète entre toutes les colonnes et l'*intent* (164 tables). Cette proportion augmente de manière significative dans ARXIV2TABLE (Wang *et al.*, 2025), où 68,5 % des *user demands* mentionnent au moins une colonne (1 341 tables), et 11,9 % décrivent explicitement l'ensemble du schéma (232 tables). Le taux moyen de correspondance atteint 37,2 %, indiquant qu'une part importante de l'information structurelle est déjà présente dans l'entrée.

Ce biais apparaît clairement dans certains exemples où l'intention ou la demande utilisateur encodent directement le schéma du tableau. Dans l'exemple 4, les expressions *design goals, constraints, problem formulaion, CSI requirement et solutions* correspondent directement aux colonnes du tableau.

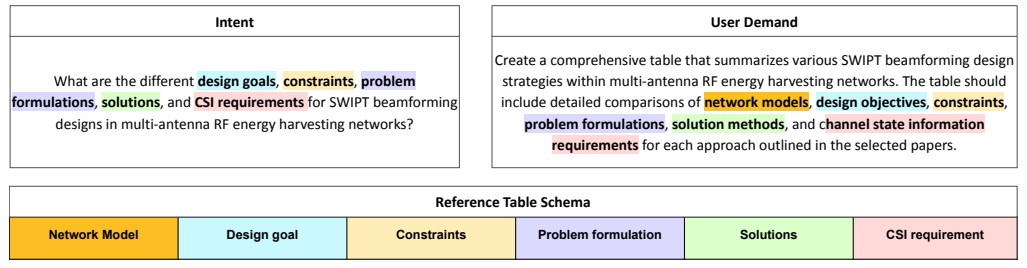

FIGURE 4 – Exemple d'*intent* et de *user demand* reconstruisant explicitement le schéma du tableau.

Ici, toutes les colonnes sont directement mentionnées dans la requête, ce qui transforme la tâche de génération en une simple instanciation plutôt qu'en une véritable inférence du schéma. Cette situation est en contradiction avec l'objectif affiché par les auteurs, qui visent à modéliser des besoins utilisateurs réalistes, où le schéma n'est généralement pas explicitement spécifié. Elle introduit également un biais dans l'évaluation, car les modèles peuvent exploiter ces indices pour améliorer artificiellement leurs performances sans réellement comprendre la structure sous-jacente des données.

Au-delà de ce problème de fuite d'information (*schema leakage*), une limite majeure réside dans la difficulté d'alignement des schémas. Comme le soulignent Newman *et al.* (2024), les colonnes sont souvent courtes, ambiguës et fortement dépendantes du contexte, ce qui rend leur appariement automatique complexe, même avec des méthodes sémantiques avancées. Par exemple, des colonnes telles que *Method*, *Approach* ou *Model* peuvent être utilisées de manière interchangeable ou, au contraire, recouvrir des distinctions subtiles selon les articles. Cette variabilité limite la fiabilité des métriques basées sur des *embeddings*, qui peuvent échouer à capturer ces nuances.

Les protocoles d'évaluation existants présentent également des limites importantes. D'une part, les métriques utilisées, telles que BERTScore, restent peu interprétables et rendent difficile l'analyse fine des erreurs. D'autre part, la validation humaine demeure partielle et souvent limitée à des sous-tâches spécifiques, comme la qualité des questions dans Wang *et al.* (2025), sans fournir une évaluation globale et systématique de la pertinence des tableaux générés. Par ailleurs, les approches actuelles reposent majoritairement sur des comparaisons avec un schéma de référence sans prendre compte des intentions ou des valeurs.

Enfin, une limite fondamentale concerne l'écart entre *similarité structurelle* et *utilité réelle*. Un tableau peut différer du tableau de référence tout en restant pertinent, voire plus informatif, par exemple en introduisant de nouvelles colonnes ou en restructurant l'information de manière plus adaptée à un besoin donné. Les méthodes d'évaluation actuelles pénalisent généralement ces variations, ce qui peut conduire à sous-estimer la qualité réelle des tableaux générés.

Ces limites ouvrent plusieurs perspectives de recherche. Une première direction consiste à concevoir des jeux de données plus larges et dans lesquels les besoins utilisateurs sont formulés sans révéler explicitement le schéma cible, afin de mieux refléter des scénarios réels. Une autre piste consiste à développer des protocoles d'évaluation capables de prendre en compte simultanément la structure, la sémantique et l'utilité des tableaux, par exemple en combinant des approches d'alignement avec des évaluations fonctionnelles basées sur des tâches d'accès à l'information. Enfin, l'intégration de validations humaines plus systématiques et le développement de métriques plus interprétables apparaissent comme des étapes essentielles pour améliorer la fiabilité et la robustesse des évaluations.

# 4   Conclusion

La génération automatique de tableaux de revue de littérature constitue une tâche à fort enjeu pour l'analyse et la structuration des travaux scientifiques. Les recherches récentes ont permis des avancées importantes, notamment à travers la création de jeux de données dédiés et le développement de premières approches de génération et d'évaluation. Toutefois, ces travaux mettent également en évidence la complexité de la tâche, en particulier au niveau du schéma, dont la variabilité dépasse largement celle des valeurs. Par ailleurs, les méthodes d'évaluation actuelles restent limitées, tant par leur dépendance à des références uniques que par leur difficulté à capturer la qualité sémantique et l'utilité réelle des tableaux. Ces constats ouvrent des perspectives de recherche importantes, notamment vers la conception de jeux de données plus larges avec des intentions plus réalistes, le développement de protocoles d'évaluation combinant structure, sémantique et utilité, ainsi que l'intégration de validations humaines et de métriques plus interprétables. Ces directions apparaissent essentielles pour progresser vers des systèmes capables de générer des tableaux réellement utiles et fiables dans des contextes de revue de littérature.

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
