# OpenReview forum: "Evaluating LLM-Generated Literature Review Tables : A Survey and Preliminary Study"
_ls2n.fr/CORIA-TALN/2026/Workshop/ARTS — ls2n CORIATALN 2026 Workshop ARTS Submission_

### Official Review · Reviewer_B3pW · 2026-04-21

**Mode De Presentation:** Poster

**Confience:**

Oui

**Decision:**

Accepté

**Relecture:**

Le papier aborde une tâche importante, à savoir l’évaluation des tableaux de revue de littérature générés par les LLM. Toutefois, bien que son objectif affiché soit de proposer un état de l’art, la couverture reste limitée à quatre jeux de données, tous dérivés d’articles issus d’ArXiv. Par ailleurs, la tâche étudiée présente de fortes similarités avec celle de la construction de leaderboards, dans la mesure où il s’agit de structurer, comparer et synthétiser des résultats scientifiques selon des critères standardisés :

Yufang Hou, Charles Jochim, Martin Gleize, Francesca Bonin, and Debasis Ganguly. 2019. Identification of Tasks, Datasets, Evaluation Metrics, and Numeric Scores for Scientific Leaderboards Construction. In Proceedings of the 57th Annual Meeting of the Association for Computational Linguistics, pages 5203–5213, Florence, Italy. Association for Computational Linguistics.

Ermakova, L. et al. (2024). Overview of the CLEF 2024 SimpleText Track. In: Goeuriot, L., et al. Experimental IR Meets Multilinguality, Multimodality, and Interaction. CLEF 2024. Lecture Notes in Computer Science, vol 14959. Springer, Cham. https://doi.org/10.1007/978-3-031-71908-0_13

Pedro Rodriguez, Joe Barrow, Alexander Hoyle, John P. Lalor, Robin Jia, and Jordan Boyd-Graber. 2021. Evaluation Examples are not Equally Informative: How should that change NLP Leaderboards?. In Proceedings of the 59th Annual Meeting of the Association for Computational Linguistics and the 11th International Joint Conference on Natural Language Processing (Volume 1: Long Papers), pages 4486–4503, Online. Association for Computational Linguistics.

**Resume:**

L’article propose un état de l’art sur l’évaluation des tableaux de revue de littérature générés par les modèles de langage de grande taille (LLM). Il présente plusieurs jeux de données récents conçus pour cette tâche, notamment ArxivDIGESTables (Newman et al., 2024), ArxivDIGESTables-with-intent et ArxivDIGESTables-Clean (Padmakumar et al., 2025), ainsi que ARXIV2TABLE (Wang et al., 2025), qui visent à structurer automatiquement des informations issues d’articles scientifiques sous forme tabulaire. Dans un second temps, le papier introduit différentes méthodes d’évaluation de ces tableaux générés. Enfin, les auteurs mettent en évidence les limites actuelles de ces méthodes.

---

### Official Review · Reviewer_Qn6m · 2026-04-26

**Mode De Presentation:** Poster

**Confience:**

Oui

**Decision:**

Accepté

**Relecture:**

Le thème de l'article est bien dans les thèmes de l'atelier; et l'article est dans l'ensemble bien rédigé et argumenté - même s'il comprend un peu des redites un peu fatales, au vu du petit nombre de travaux antérieurs sur le sujet, dont on se demande s'il est vraiment mûr pour être soit même l'objet d'une revue de littérature. Il manque également peut-être un peu de regard critique sur la tâche elle-même, dont on se demande comment la définir correctement en l'absence d'une définition claire sur (a) la revue de littérature dans laquelle elle doit figurer, et qui conditionne les articles qui sont sélectionnés puis analysés; (b) de l'intention de la personne qui rédige ladite revue, laquelle pourra contenir de multiples tables, chacune couvrant un des aspects du sujet analysé.

Les auteurs pourraient aussi discuter un peu plus des lacunes des principaux jeux  de donnée utilisés, qui reposent essentiellement sur des données collectées sur arXiv - il existe des sources fiables qui se consacrent à des études de littérature - ACM Comput. Survey pour l'informatique, mais il existe certainement des équivalents dans d'autres domaines scientifiques.

**Resume:**

Cet article s'intéresse à la tâche d'extraction d'informations structurées (sous une forme tabulaire) à partir d'un ensemble de documents scientifiques dans le but d'alimenter un revue de littérature. Les auteurs discutent essentiellement trois aspects: les méthodes de génération de tables -- à la fois l'induction d'une structure (quelles en sont les colonnes, quelles en sont les lignes, et le remplissage des cellules); les données pour apprendre à produire de telles tables; enfin les questions relatives à l'évaluation de tables produites automatiquement.

---

### Decision · Program_Chairs · 2026-05-07

Accept (Poster)